2LE-BO-DeepTrade: an integrated deep learning framework for stock price prediction

Akşehir Zinnet Duygu duygu.aksehir@bil.omu.edu.tr
Kılıç Erdal
Department of Computer Engineering, Ondokuz Mayis University , Samsun , Turkey
Cunkas Mehmet
Electronic publication date: 2025 Aug 8
Publication date: 2025
Volume: 11
Electronic Location ID: e3107
Received 2025 Apr 17; Accepted 2025 Jul 16
Copyright: © 2025 Akşehir and Kılıç
Copyright year: 2025
Copyright holder: Akşehir and Kılıç
License: This is an open access article distributed under the terms of the Creative Commons Attribution License, which permits unrestricted use, distribution, reproduction and adaptation in any medium and for any purpose provided that it is properly attributed. For attribution, the original author(s), title, publication source (PeerJ Computer Science) and either DOI or URL of the article must be cited.
License URL: https://creativecommons.org/licenses/by/4.0/

Keywords: Noise reduction, Deep learning, Mode decomposition, 2LE-ICEEMDAN, Stock price prediction, Trading strategy

Funding: Ondokuz Mayıs University BAP PYO.MUH.1904.23.002 This work was supported by Ondokuz Mayıs University BAP under grant PYO.MUH.1904.23.002. The funders had no role in study design, data collection and analysis, decision to publish, or preparation of the manuscript.

==============================
This study presents a novel, integrated deep-learning framework named 2LE-BO-DeepTrade for stock closing price prediction. This framework combines 2LE-ICEEMDAN denoising, deep learning models tuned with Bayesian optimization, and a piecewise linear representation (PLR)-based trading strategy. The framework utilizes the model that provides the highest accuracy among optimized long short-term memory (LSTM), long short term memory with batch normalization (LSTM-BN), and gated recurrent unit (GRU) models on data preprocessed with the 2LE-ICEEMDAN denoising method. The model’s performance is comprehensively evaluated using both statistical metrics and a PLR-based trading strategy specifically developed for this study. Experimental studies were conducted on AKBNK, MGROS, KCHOL, THYAO, and ULKER stocks, which are traded on Borsa Istanbul and represent different sectors. During the denoising phase, noise in the stock prices was successfully removed, and noiseless intrinsic mode functions (IMFs) were obtained. The optimal model and hyperparameters for each IMF component were determined using Bayesian optimization, significantly improving prediction accuracy. The model within this framework, characterized by its optimized yet simple structure, demonstrated superior predictive performance compared to the more complex ICE2DE-MDL model in the literature. When compared to ICE2DE-MDL, the 2LE-BO-DeepTrade model, across all tested stocks, reduced the average root mean square error (RMSE) value by 94.4%, the average mean absolute error (MAE) value by 93.6%, and the average mean absolute percentage error (MAPE) value by 37.4% while increasing the average R2 value by 1.1%. Furthermore, the PLR-based trading strategy, specifically developed for this study, generated “Buy” and “Sell” signals, exhibiting a remarkably superior financial performance to a passive investment strategy. Across all considered stocks, the PLR-based strategy yielded, on average, 66 times more profit than the passive approach. These findings substantiate that the proposed integrated deep learning-based stock forecasting framework can significantly enhance the accuracy of stock market predictions and the returns of trading strategies.

Introduction

The stock market plays a vital role in the global economy, offering investors the potential for substantial returns while also engendering significant risks. Therefore, accurate prediction of the price movements of individual stocks and the indices used to measure the stock market’s performance is critical for monitoring market volatility and developing investment strategies. However, the stock market, influenced by various dynamic factors such as central bank policy adjustments, changes in monetary policy, geopolitical instability, company policies, investor decisions, and macroeconomic factors, exhibits a complex and unpredictable nature (Rouf et al., 2021). This makes it difficult to predict market movements, according to random walk theory (Fama, 1970).

Stock forecasting is essential for overcoming this unpredictability, which poses significant challenges for investors. Stock forecasting is an interdisciplinary field of study that aims to predict the future price of stock indices or individual stocks. In this field, both traditional and modern methods are employed (Shah, Vaidya & Shah, 2022). While traditional methods rely on the assumption of linear and stationary data structures, modern machine learning and deep learning techniques are better able to model nonlinear relationships and complex data structures (Liu et al., 2022; Shi et al., 2021). In particular, deep learning models have achieved significant success in stock market forecasting due to their multi-layered structures and ability to extract meaningful patterns from large datasets (Akşehir & Kılıç, 2024a; Kanwal et al., 2022; Fisichella & Garolla, 2021; Balasubramanian et al., 2024). However, significant obstacles hinder fully realizing the potential of deep learning (DL) models in financial time series forecasting. A fundamental problem is the inherently high level of noise present in financial data. This noise obscures the underlying patterns, making it difficult for DL models to learn them, thereby degrading prediction performance. A notable gap in the existing literature is the limited systematic integration of advanced signal processing techniques—capable of effectively addressing this noise issue and, particularly, of precisely decomposing different frequency components—into deep learning processes. Another critical problem is the extreme sensitivity of DL models’ performance to architectural choices and hyperparameter settings. This highlights the lack of integrated optimization strategies capable of automatically and efficiently determining the most suitable DL model and its hyperparameters for each signal component, especially when multiple signal components with varying characteristics are obtained after denoising. Finally, a common deficiency is apparent in evaluating the practical value of developed forecasting models; many studies assess model performance solely based on statistical error metrics, yet they often neglect how these metrics reflect the model’s financial viability and profitability in real-world trading scenarios.

The primary motivation of this study is to address these identified problems and gaps in the literature by developing an integrated deep learning framework for stock price forecasting that can produce statistically accurate and practically applicable results. Accordingly, we propose a novel hybrid framework named 2LE-BO-DeepTrade. The proposed framework aims to fill these gaps: (1) It utilizes the 2LE-ICEEMDAN method, which is based on two-level mode decomposition and entropy concepts, to effectively decompose noise in financial time series and obtain cleaner signal components. (2) It applies Bayesian Optimization to automatically optimize the most suitable deep learning model and its hyperparameters for each decomposed IMF component. (3) It evaluates the model’s predictive performance not only through statistical metrics but also in terms of its financial return potential via our newly proposed trading strategy, thereby demonstrating its practical applicability. This integrated approach, by combining noise reduction, model optimization, and comprehensive performance evaluation within a cohesive framework, aims to enhance the accuracy of stock market predictions and the effectiveness of investment strategies.

Related work

Stock forecasting has long been an area of interest for researchers due to the complex structure and dynamic nature of financial markets. Studies in this field primarily encompass data preprocessing, modeling, and evaluation processes. The data preprocessing stage, in particular, stands out as a critical step that directly affects prediction accuracy. Financial data is often incomplete and noisy; therefore, preprocessing the data to remove noise is paramount for improving model performance. The literature includes many studies on preprocessing steps, such as completing missing data, detecting outliers, and noise reduction techniques. In this context, Lv et al. (2022) stated that traditional methods are insufficient for stock index prediction and that noise in financial time series negatively impacts prediction performance. To address this issue, they proposed a hybrid method called CEEMDAN-DAE-LSTM. The method decomposes closing prices into IMFs using Complete Ensemble Empirical Mode Decomposition with Adaptive Noise (CEEMDAN), removes redundant data by extracting features with a deep autoencoder (DAE), and uses long short-term memory (LSTM) for prediction. Test results indicate that CEEMDAN-DAE-LSTM provides higher accuracy compared to other methods. In another study, He, Siu & Si (2023) noted that deep learning models are widely used in stock direction prediction, but the limited historical price data for newly listed stocks can reduce performance. To overcome this problem, they proposed an instance-based deep transfer learning model with an attention mechanism (IDTLA). IDTLA enables effective learning with insufficient data and is trained with LSTM model. Similarly, Qi, Ren & Su (2023) developed a hybrid model based on CEEMDAN, wavelet transform, and gated recurrent unit (GRU) for stock index prediction. While CEEMDAN decomposes index data into IMFs, the wavelet transform removes noise in high-frequency IMFs. The GRU model was trained with the denoised data for prediction and tested on the S&P 500 and CSI 300 indices. The results showed that the model outperformed traditional methods such as GRU, LSTM, autoregressive integrated moving average (ARIMA), convolutional neural network-bidirectional long short-term memory (CNN-BiLSTM), and artificial neural network (ANN). Zhao & Yang (2023) noted that machine learning methods are used in stock price direction prediction, but that price data itself makes prediction difficult. To address this issue, they proposed the self-attention based difference long short-term memory (SA-DLSTM) framework, combining sentiment analysis, a denoising autoencoder, and LSTM model. A sentiment index was constructed from forum texts associated with Hang Seng Index (HSI) index price data, features were extracted and noise was reduced using a denoising autoencoder, and direction prediction was performed using LSTM. Compared with traditional methods, the model demonstrated high accuracy and profitability, identifying suitable trading points. Safari & Ghaemi (2025) proposed NeuroFuzzyMan predictor, a novel hybrid model that integrates neuro-fuzzy logic and bidirectional long short-term memory (BiLSTM) networks for financial time series forecasting. This model captures uncertainty in financial data with fuzzy logic and learns long-term dependencies and complex patterns with BiLSTM networks. The model was tested in JPMorgan, Amazon and Tesla stocks, and the results demonstrate the accuracy and robustness of NeuroFuzzyMan using various performance indicators. Cui et al. (2023) stated that stock index prediction is challenging due to the non-linear structure and noise present in stock index data. To overcome this problem, they focused on hybrid models, defining three categories: pure machine learning/deep learning, time series-deep learning, and decomposition-deep learning hybrid models. They proposed the McVCsB model, which combines variational mode decomposition (VMD), convolutional block attention module (CBAM), and BiLSTM methods, for stock index prediction. VMD was used for noise reduction, CBAM for extracting deep features, and BiLSTM for prediction. Experiments on the SSCI, STI, FTSE, and S&P 500 indices showed that the model exhibited superior performance and that its components contributed positively to the prediction. Wang, Liu & Jiang (2024) proposed an interval-valued decomposition integration model for predicting stock indices’ high and low prices, emphasizing that interval-value prediction is more suitable than point-value prediction. The model involves three components: interval multiscale decomposition, multifactor analysis, and nonlinear prediction using intelligent optimization. Stock prices are first decomposed into interval trends and residuals using the interval variational mode decomposition with feedback mechanism (FIVMD). Multifactor analysis examines external factors such as global equity markets and commodity prices, while LightGBM identifies key influencing factors. Dimensionality reduction is done via an autoencoder to reduce complexity. Finally, LSTM, optimized with the whale optimization algorithm (WOA), predicts interval trends and residual series separately. The model was tested on the Shenzhen, Shanghai, and China Securities 100 indices, showing superior performance compared to nine other models. An et al. (2024) proposed the HFSLSMR-LSTM model, combining hierarchical feature selection with local shuffling (HFSLS) and model reweighting (MR) based on LSTM. It was evaluated on stock market indices and individual stock prices using four years of closing price data from YahooFinance for eight indices and 30 DJIA stocks. A total of 158 technical indicators were calculated, and LSTM was used for forecasting after feature selection. The model outperformed six other models, demonstrating its effectiveness in predicting stock prices in volatile markets. Li et al. (2024) proposed the Boruta-MICAN-BSVR method, which combines Boruta feature selection and adaptive noise reduction techniques, for stock index prediction. With this method, important features were selected from 31 technical indicators using Boruta, followed by applying the MIC-ICEEMDAN noise reduction method. After denoising, prediction was performed using support vector regression (SVR) and the brain storm optimization algorithm (BSO). The method was tested on Chinese stock indices (SSEC, SZSE, CSI 500), and compared to eight other methods, it demonstrated lower error rates and higher accuracy. Akşehir & Kılıç (2024c) proposed the 2LE-CEEMDAN method for noise reduction in stock market data, addressing the limitations of existing techniques. A prediction model combining 2LE-CEEMDAN, LSTM, and SVR was used to forecast the next day’s closing prices for indices like S&P 500, DAX, DJI, and SSE. The model outperformed others, proving the effectiveness of this denoising approach. In another study, Akşehir & Kılıç (2024b) introduced the ICE2DE-MDL hybrid model, which applies individualized models to decomposed sub-series, showing superior performance in stock price prediction and highlighting the importance of individualized modeling for noise reduction.

While addressing noise and missing data during the data preprocessing stage is crucial to improve model performance, modeling stock data’s complex and non-linear structure is just as decisive as data quality. However, the complex and non-linear nature of stock data also presents significant challenges in the modeling phase, creating situations that cannot be overcome solely by data preprocessing techniques. Therefore, the modeling methods used to increase prediction success must also be carefully selected. In this context, statistical methods were initially preferred better to model the non-linear structure of stock market data. Although models such as autoregressive moving average (ARMA) (Rather, Agarwal & Sastry, 2015), autoregressive integrated moving average (ARIMA) (Babu & Reddy, 2015), and generalized autoregressive conditional heteroscedastic (GARCH) (Ariyo, Adewumi & Ayo, 2014) have yielded successful results on less complex and low-noise datasets, the complex and non-linear nature of stock market data has caused these methods to be insufficient (Ji, Liew & Yang, 2021; Zhang & Lei, 2022). These limitations have led researchers to more advanced methods, resulting in the proposal of various machine learning and deep learning techniques (Shi et al., 2021). Studies on such innovative methods based on machine learning and deep learning have gradually increased in the literature. For instance, Albahli et al. (2023) proposed a stock trend prediction model to assist investors. The model used historical stock prices and 18 technical indicators, which were reduced in dimensionality using an autoencoder. The reduced features and price data were input into the DenseNet-41 model, which was optimized by reducing the number of Dense blocks from four to three to prevent overfitting. Tested on 10 S&P 500 stocks, the model outperformed other machine learning and deep learning methods, with the DenseNet-41’s feature extraction capability being key to its success. Yao, Zhang & Zhao (2023) introduced the MEMD-TCN hybrid model to address challenges in predicting stock market indexes. The model combines multivariate empirical mode decomposition (MEMD) with temporal convolutional networks (TCN). Stock index data, including various prices and trading volume, were decomposed using MEMD, and the subseries were input into the TCN for prediction. Evaluated on seven stock indexes globally, the model outperformed 14 other models in prediction accuracy, with the combination of MEMD and TCN significantly improving performance. Md et al. (2023) proposed the MLS-LSTM model for stock price prediction, consisting of three sequential LSTM layers and one dense layer. The model used normalized time series data to capture the relationship between past and future prices. Tested on Samsung’s 5-year closing price data, MLS-LSTM outperformed several models, including multi-layer perceptron (MLP), support vector machine (SVM), linear regression, recurrent neural network (RNN), single-layer LSTM, and convolutional neural network (CNN), with an average prediction error of around two percent. Using multiple LSTM layers helped prevent overfitting and enhanced performance compared to single-layer models. Wang & Zhu (2023) proposed a stock trend prediction model integrating sentiment analysis and dual classifier coupling to address the impact of investor-generated textual content on stock behavior. They introduced a sentiment index weighted by reading volume and adjusted for time decay using an exponential moving average. The model combined CNN and SVM for dual classifier coupling. Tested on the SSE 50 and CSI 300 stock indexes, it outperformed models like MLP, SVM, LSTM, and hybrid models (CNN-LSTM, CNN-SGD, CNN-BiLSTM), showing enhanced prediction accuracy with the CNN-SVM model. Yu et al. (2023) introduced the GVMD-Q-DBN-LSTM-GRU hybrid model to predict stock market indices’ realized volatility (RV) with high accuracy. The model employs the gray wolf optimization algorithm and VMD to decompose RV sequences into IMFs. Each IMF is trained using deep belief networks (DBN), LSTM, and GRU, and results are integrated with a Q-learning algorithm to determine optimal weights. Tested on indices like the Shanghai Stock Exchange Composite, FTSE, and S&P 500 using 5-minute data, the model outperformed ten other prediction models based on MAE, MSE, and other performance metrics. Chen, Wu & Wu (2022) reviewed the limitations of deep learning-based models for stock prediction, noting that many studies focus on single-stock data, which can undermine long-term prediction accuracy. The KD-LSTM model was proposed to address this, incorporating a clustering approach using k-means and Dynamic Time Warping (DTW) to group 16 bank stocks with similar price trends on the Chinese stock market. Data from four stocks within the same cluster were used to train the LSTM model. The experimental results showed that this clustering approach enhanced the model’s prediction performance. Yu et al. (2025) proposed a hybrid model (GA-VMD-TCN) based on VMD and TCN, optimized with a genetic algorithm (GA), for multi-step forecasting of stock indices. The model decomposes the stock index into intrinsic mode functions using VMD optimized with GA, and performs multi-step prediction with TCN for each IMF. Experimental results show that GA-VMD-TCN provides better prediction accuracy and stability compared to other methods. Mehtarizadeh et al. (2025) proposed a hybrid model that uses a LSTM network modified with the Sine-Cosine Algorithm (SCA), ARIMA, and GARCH models for stock price prediction. This model aims to manage volatility by effectively capturing linear and non-linear patterns. In the study, experiments with data from 12 different stocks showed that daily predictions outperformed traditional LSTM and other hybrid models by 83.37%, 84.05%, and 55.8%, respectively. Furthermore, the LSTM-SCA-ARIMA-GARCH model demonstrates a significant superiority over other alternative methods.

Considering the methods used in the modeling process and their success, the next critical step is to evaluate the effectiveness and adaptability of these models, thereby measuring prediction accuracy and reliability. Statistical metrics such as MSE, MAE, and R-squared are frequently used to assess prediction accuracy in the literature. Some studies also utilize performance evaluation methods integrated with trading strategies. In this context, Kim et al. (2023) aimed to predict the direction of the KOSPI 200 index while identifying key features used by the model. The study used 30-min KOSPI 200 data and direction labeling based on consecutive data points, considering 16 technical indicators. A CNN model was trained on 16 × 16 image datasets derived from 16-period values. The gradient-weighted class activation mapping (Grad-CAM) method was applied to identify influential features. The results showed that based on Grad-CAM insights, the proposed trading strategy outperformed alternative strategies, including buy-and-hold and those based on common technical indicators. Fang et al. (2023) addressed the limitations of LSTM models in long-term financial time series predictions due to their non-linear and non-stationary nature. They proposed a modified adaptive LSTM model based on an LSTM-BN network with a multi-path parallel structure. The model incorporated batch normalization and dropout layers to prevent overfitting and used a parallel structure with multiple LSTM-BN networks to enhance prediction stability. A voting method determined the final prediction, and an adaptive cross-entropy loss function was introduced for sharp change points in financial data. The model, tested on the S&P 500, CSI 300, and SSE 180 indices, outperformed five other models in terms of accuracy and return. Shah et al. (2022) proposed a hybrid deep learning model combining CNN and LSTM to predict the Nifty 50 stock index’s closing price. The model utilized 48 features, including raw price data, technical indicators, foreign indices, exchange rates, and commodity prices. CNN was used to extract high-level features, while LSTM predicted the next day’s value. The model achieved a mean absolute percentage error (MAPE) of 2.54% on the training dataset, outperforming similar studies. Over 10 years, the CNN-LSTM model yielded a return of 342%, compared to 107% from the buy-and-hold strategy, demonstrating superior performance.

Significant advancements have been recorded in stock forecasting, with numerous studies proposing various methodologies. These studies are summarized in Table 1 concerning their critical approaches, including noise reduction, hyperparameter optimization, the integration of trading strategies, and the utilization of individual modeling strategies. As illustrated in Table 1, while studies such as (Lv et al., 2022) demonstrate the value of employing decomposition techniques for noise reduction, a prevalent limitation in many such works is the omission of systematic hyperparameter optimization for their prediction models or lack of validation through realistic trading strategies. This oversight constitutes a deficiency in pursuing practical and optimal solutions, a core aspect of our problem statement. Similarly, although hyperparameter optimization, as seen in Wang, Liu & Jiang (2024), or trading strategies, as in Zhao & Yang (2023), are encountered in the literature, these approaches often lack holistic integration. The 2LE-BO-DeepTrade framework proposed in this study aims to address these literature gaps. Indeed, as highlighted in Table 1, this framework uniquely offers a comprehensive approach, seldom found in existing literature, by simultaneously integrating (i) advanced and individualized noise reduction with 2LE-ICEEMDAN, (ii) robust hyperparameter optimization, proven effective across multiple deep learning architectures, via Bayesian optimization, and (iii) validation through a concrete profit-loss analysis based on a proposed trading strategy.

Table 1 Summary and comparison of related works.

Studies	Noise reduction	Hyperparameter optimization	Trading strategy	Individual modelling strategy	
Lv et al. (2022)	✓ (CEEMDAN-DAE)	✗	✗	✓	
He, Siu & Si (2023)	✗	✗	✗	✗	
Qi, Ren & Su (2023)	✓ (CEEMDAN-Wavelet transform)	✗	✗	✗	
Zhao & Yang (2023)	✗	✗	✓	✗	
Safari & Ghaemi (2025)	✗	✗	✗	✗	
Cui et al. (2023)	✓ (VMD)	✗	✗	✗	
Wang, Liu & Jiang (2024)	✗	✓ (Whale optimization algorithm)	✗	✗	
An et al. (2024)	✗	✗	✗	✗	
Li et al. (2024)	✓ (MIC-ICEEMDAN)	✓ (Brain storm optimization algorithm)	✗	✗	
Akşehir & Kılıç (2024c)	✓ (2LE-CEEMDAN)	✗	✗	✓	
Akşehir & Kılıç (2024b)	✓ (2LE-ICEEMDAN)	✗	✗	✓	
Albahli et al. (2023)	✗	✗	✗	✗	
Yao, Zhang & Zhao (2023)	✗	✗	✗	✓	
Md et al. (2023)	✗	✗	✗	✗	
Wang & Zhu (2023)	✗	✗	✗	✗	
Yu et al. (2023)	✗	✓ (Gray wolf optimization algorithm)	✗	✓	
Chen, Wu & Wu (2022)	✗	✗	✗	✗	
Yu et al. (2025)	✗	✓ (Genetic algorithm)	✗	✓	
Mehtarizadeh et al. (2025)	✗	✓ (Sine-cosine algorithm)	✗	✗	
Kim et al. (2023)	✗	✗	✓	✗	
Fang et al. (2023)	✗	✗	✓	✓	
Shah et al. (2022)	✗	✗	✓	✗	
2LE-BO-DeepTrade	✓ (2LE-ICEEMDAN)	✓ (Bayesian optimization algorithm)	✓	✓	

Motivation and contributions

The increasing volatility and complexity of financial markets underscore the critical need for reliable and precise forecasting models for investors. Traditional approaches address the stock prediction process by focusing separately on stages such as data preprocessing, denoising, model optimization, and trading strategy development. This fragmented approach prevents a holistic evaluation of the process and does not provide a viable framework for investors. The primary motivation of this study, therefore, is to overcome the limitations of such fragmented approaches by developing a complete, data-driven forecasting framework. This framework aims to integrate modeling and decision-making, encompassing the entire process from data preprocessing to trading strategy development, ultimately to enhance investor profitability. Building on this motivation, this study makes the following key contributions: Integrated framework: To the best of our knowledge, our proposed 2LE-BO-DeepTrade framework represents one of the first end-to-end integrated approaches for stock price prediction. Its novelty lies in systematically combining advanced denoising, robust Bayesian-optimized hybrid deep learning, and a practical PLR-based trading strategy within a unified structure. This offers a more coherent, adaptable, and effective solution compared to traditional approaches, where data preprocessing, modeling, and trading strategy development stages are addressed independently.

Enhanced denoising: The 2LE-ICEEMDAN method, previously proposed in Akşehir & Kılıç (2024b) by the authors, effectively reduces noise in stock price data, enabling deep learning models to be trained with cleaner and more meaningful data.

Optimized deep learning modeling: A modeling approach consisting of LSTM, LSTM-BN, and GRU models, with hyperparameters optimally tuned using Bayesian optimization, was employed. This approach maximized prediction accuracy by selecting the most suitable model for each IMF.

Proven financial returns: The developed PLR-based trading strategy, by directly converting the model’s predictions into investment decisions, achieved returns approximately 66 times higher than the traditional Buy-and-Hold (passive) strategy on average for the considered stocks. This clearly demonstrates that the model is not only statistically successful but also practically and financially valuable.

Organization

The remainder of this article is organized as follows: The “Methodology” provides a detailed description of all the methods used in the 2LE-BO-DeepTrade forecasting framework, including the denoising method, deep learning models, hyperparameter optimization with Bayesian optimization, and the PLR-based trading strategy. The “2LE-BO-DeepTrade Framework” explains the overall structure and operation of the proposed forecasting framework step-by-step, clearly outlining the relationships between the different components of the framework. The “Experimental Settings” details how the experimental studies were conducted, introducing the dataset used, the search space for hyperparameter optimization. The “Result and Discussion” presents and discusses the results of the experimental studies comprehensively, evaluating the prediction results obtained for different stocks in terms of both statistical metrics and returns from the PLR-based trading strategy; furthermore, the performance of the proposed model is compared with methods in the literature, the strengths and weaknesses of the study are analyzed, and graphical representations of the results are provided. Finally, the “Conclusion and Future Works” summarizes the main findings and conclusions of the study, highlights the contributions to the literature, offers suggestions for future work, and discusses the study’s limitations.

Methodology

This section introduces the methodology of 2LE-BO-DeepTrade, the integrated deep learning-based forecasting framework developed for stock price prediction. Data preprocessing techniques, deep learning models, hyperparameter optimization with Bayesian Optimization, commonly used statistical performance metrics, and the PLR-based trading strategy, one of the study’s key contributions, are detailed.

Data preprocessing and denoising

Reducing noise in financial time series significantly improves the performance of forecasting models, and methods such as Fourier transform, wavelet transform, and signal decomposition methods are widely used in the literature (Akşehir & Kılıç, 2024a). However, techniques like Fourier and wavelet transforms are insufficient in analyzing financial data’s dynamic and complex structure. These limitations make mode decomposition methods a more effective alternative.

Mode decomposition methods reveal the intrinsic structure of data by decomposing time series into different frequency components IMFs and allow each component to be analyzed separately. Advanced versions, such as CEEMDAN and Improved CEEMDAN (ICEEMDAN), exhibit superior performance in denoising financial time series due to their more precise decomposition capabilities. In this study, the 2LE-ICEEMDAN method was used to denoise financial time series, previously proposed in our work (Akşehir & Kılıç, 2024b). This method is based on a two-level ICEEMDAN mode decomposition and an entropy-based approach, and its effectiveness has been demonstrated in previous studies. The implementation steps of this method are detailed below: Step 1: The time series data is taken as input.

Step 2: First-level decomposition is performed on the input data using the ICEEMDAN method, and IMFs are obtained.

Step 3: Sample and approximate entropy values are calculated for each IMF obtained from the decomposition.

Step 4: The sample entropy ratio and approximate entropy ratio are determined by dividing the entropy of each IMF by the total entropy.

Step 5: The IMFs are divided into two groups: noisy and noiseless components, according to the following rules: – If an IMF’s sample entropy ratio or approximate entropy ratio of greater than 20%, this IMF is classified as a high-frequency (noisy) component.

– Otherwise, the IMF is considered a noiseless component.

Step 6: The noisy components identified in the first-level decomposition are summed, and a second decomposition is performed using the ICEEMDAN method.

Step 7: Steps 3–5 are repeated, and the IMFs obtained from the second decomposition are also classified as noisy and noiseless components.

Step 8: Finally, the noiseless IMFs obtained in the first and second decomposition steps are summed to create the denoised time series.

Deep learning models and Bayesian optimization

Within the scope of this study, LSTM, GRU, and LSTM-BN were chosen as deep learning models due to their ability to model complex and long-term dependencies in financial time series. The Bayesian Optimization Algorithm was used for hyperparameter optimization of these models. In this section, all methods used for modeling are explained in detail.

Long short term memory and long short term memory with batch normalization

Recurrent neural networks (RNNs) can process sequential data and model relationships between data points (Elman, 1990). However, RNNs can face challenges such as the “long-term dependency” and “vanishing gradient” problems (Staudemeyer & Morris, 2019; Young et al., 2018). To overcome these issues, LSTM networks, developed by Hochreiter and Schmidhuber, stand out, particularly with their ability to learn long-term dependencies (Hochreiter, 1997).

Unlike traditional RNNs, LSTM has a more complex structure, and thanks to this structure, it has a better capacity for learning and establishing relationships. While the RNN architecture consists of a single layer containing only a tanh layer, the LSTM architecture has a more complex structure consisting of four consecutive layers (Hochreiter, 1997). As seen in Fig. 1, these layers consist of the cell state, output gate, input gate, and forget gate. LSTM enables information to be remembered for long periods and is updated at each time step through the connections between these layers.

Figure 1 LSTM architecture.

Equations (1)–(6) show how the LSTM unit works at time step t for an input vector X. In these equations, it represents the input gate, ft the forget gate, ot the output gate, Ct the memory cell, ht the hidden state, W the weight matrices, b the bias vector, and σ the sigmoid activation function.

(1) forgetl=σ(Wforget⋅[hiddenl−1,Xl]+bforget)

(2) inputl=σ(Winput⋅[hiddenl−1,Xl]+binput)

(3) outputl=σ(Woutput⋅[hiddenl−1,Xl]+boutput)

(4) Celll~=tanh(Wcell⋅[hiddenl−1,Xl]+bcell)

(5) Celll=forgetl⊙Celll−1+inputl⊙Celll~

(6) hiddenl=outputl⊙tanh(Celll)

Fang et al. (2023) proposed the LSTM-BN method for predicting financial time series movements was developed by adding a batch normalization (BN) layer to the traditional LSTM architecture. This additional layer accelerates the training process and enables the model to operate more stably, thereby improving the network’s overall performance.

Gated recurrent unit

GRU, a type of RNN, was developed by Cho et al. (2014) as an alternative to LSTM to address the vanishing gradient problem and capture long-term dependencies in sequential data. As shown in Fig. 2, GRU combines the memory cell in LSTM with the hidden state, reducing the number of parameters and thus offering a more simplified architecture.

Figure 2 GRU architecture.

GRU fundamentally consists of update ( updatel) and reset ( resetl) gates. These gates control the flow of information within the unit, enabling the hidden state to be updated selectively. The mathematical definition of the GRU cell is given in Eqs. (7)–(10). These equations show how the hidden state is calculated and updated at each time step. Here, σ represents the sigmoid activation function, and tanh represents the hyperbolic tangent activation function. The ⊙ operator denotes element-wise multiplication, W denotes the weight matrix, and b denotes the bias vector.

(7) updatel=σ(Wupdate⋅[hiddenl−1,Xl]+bupdate)

(8) resetl=σ(Wreset⋅[hiddenl−1,Xl]+breset)

(9) hiddenl~=tanh(Whidden⋅[resetl⊙hiddenl−1,Xl]+bhidden)

(10) hiddenl=(1-updatel)⊙hiddenl−1+updatel⊙hiddenl~.

Bayesian optimization

Various hyperparameter optimization techniques exist in the literature; commonly used methods such as grid search, random search, and Bayesian optimization stand out. The efficiency of the grid search method decreases as the number of hyperparameters increases. The exponential increase in the number of required evaluations with each additional parameter significantly increases the computational time complexity (Yue et al., 2022). On the other hand, the random search technique uses randomly sampled parameter combinations based on a statistical distribution. However, this approach is insufficient in finding optimal hyperparameter points for some complex models (Yang, Li & Xun, 2019). In contrast to the grid and random search methods, Bayesian optimization determines new parameter values based on past data (Frazier, 2018). This approach provides a more efficient optimization process by making more effective choices, considering the results of previous trials.

Bayesian optimization comprises two fundamental components: Surrogate model: This model is used to estimate the probability distribution of the objective function. Based on past evaluation results, this model predicts the possible values of the objective function. Although many surrogate models exist, the Gaussian process is among the most frequently preferred models (Yu & Zhu, 2020).

Acquisition function: In Bayesian optimization, the acquisition function is used to determine which points in the search space should be evaluated. This function predicts which hyperparameter combinations are likely to be the most optimal, based on past data. The acquisition function serves two primary purposes: first, to provide exploitation to improve the accuracy of the current model, and second, to explore unexplored areas.

The Bayesian optimization algorithm is given in Algorithm 1. This algorithm takes as input the black-box function to be optimized ( f), the surrogate model that estimates the objective function ( f^), and the parameter values to be tried (X). At the beginning of the process, a set (H) is created to store the data from previous trials. Using the current surrogate model, the parameter set ( xn) that yields the highest expected output f(xn) is selected. The actual output ( yn=f(xn)) is obtained for this selected value. The obtained pairs xn and yn are added to the set H to update the past evaluations. This process continues until a specified stopping criterion is met (Cho et al., 2020).

Algorithm 1 Bayesian optimization.

1: Input: Black-box function f, BO algorithm f^, parameter space X	
2: H←⊘               ▹ Initialize history set	
3: n←1                   ▹Initialize iteration counter	
4: While true do	
5:     xn←arg⁡maxx∈X f^(x;H)    ▹Find the next parameter set to evaluate	
6:     yn←f(xn)  ▹Evaluate the black-box function	
7:     H←H∪{(xn,yn)}           ▹Update history	
8:    Check stopping criteria	
9:     n←n+1	
10: end while	

For hyperparameter optimization of the deep learning models in this study, Bayesian optimization utilizing a Gaussian process as a surrogate model was employed. The objective function for this optimization process is defined as a metric that measures the performance of the deep learning model, given a specific hyperparameter set, on the validation dataset. Specifically, this objective function is set to minimize the RMSE value. This choice aims to enhance the model’s generalization ability and prevent overfitting, as the optimization is performed on the validation set rather than directly on the test set. The hyperparameter set yielding the lowest root mean squared error (RMSE) is considered the best model configuration for that particular IMF component.

Model performance evaluation and validation

This section describes the statistical metrics (MSE, MAE, RMSE, R-squared) and the PLR-based trading strategy used to evaluate the model’s prediction performance and financial returns.

Statistical performance metrics

In this study, four common metrics were used to evaluate the performance of the proposed forecasting statistically: RMSE, mean absolute error (MAE), mean absolute percentage error (MAPE), and R-squared. The mathematical formulations for calculating these metrics are given in Table 2. In these mathematical formulations, n signifies the number of data points, P represents the predicted value, A indicates the actual value, and A¯ denotes the arithmetic mean of the actual values.

Table 2 Statistical metrics.

Metrics	Formulations	
RMSE	1n∑i=1n(Pi−Ai)2	
MAE	1n∑i=1n|Pi−Ai|	
MAPE	1n∑i=1n|Pi−AiAi|	
R2	1−∑i=1n(Pi−Ai)2∑i=1n(Pi−A¯i)2	

PLR-based trading strategy

In addition to statistical metrics, this study proposes a financial evaluation approach based on PLR to assess the performance of stock prediction models. This proposed approach aims to provide investors with a more effective decision-making mechanism, also considering the model’s impact on financial outcomes.

The PLR method divides the financial time series into multiple segments and creates separate linear regression models for each segment. This segmentation highlights sudden changes in the data (e.g., trend reversals), allowing for the analysis of linear trends in different periods (Keogh et al., 2001).

The PLR algorithm includes the following steps for analyzing financial time series data: Segmentation: The financial time series data is divided into specific segments. Each segment consists of a specific range of data points, and its price movements are modeled using linear regression. This segmentation is used to reflect changes in market conditions more accurately.

Linear regression calculation: Within each segment, a linear regression model is applied to determine the trends in price movements. This model allows the prices in the segment to be expressed by a linear equation ( y=mx+b). Here, m represents the slope, in other words, the direction of market movements, and b represents the segment’s starting point.

Detection of slope changes: Changes in slope between different segments generate potential buy and sell signals. A change in a segment’s slope from negative to positive indicates that prices have entered an upward trend, and is typically interpreted as a ‘Buy’ signal. Similarly, a change in a segment’s slope from positive to negative indicates that prices have entered a downward trend, and is typically interpreted as a ‘Sell’ signal.

Benchmark model

The ICE2DE-MDL (Akşehir & Kılıç, 2024b) prediction model was selected to compare the performance of the proposed forecasting model with the current study in the literature. The existing ICE2DE-MDL hybrid model in the literature adopts a multi-step approach for predicting stock closing prices. In this model, initially, the 2LE-ICEEMDAN denoising technique is applied to the raw price data to obtain a set of denoised IMFs. Subsequently, for each IMF, both traditional machine learning (SVR) and deep learning models (such as LSTM, LSTM-BN, and GRU) are trained separately. A critical aspect of this process is the manual determination of the models’ hyperparameters. Post-training, the model exhibiting the lowest error metrics (evaluated using traditional statistical criteria like RMSE, MAE, MAPE, and R-squared) is selected as the “best model” for that specific IMF. The final prediction is generated by combining the prediction results from these IMF-specific best models, and the overall performance of the model is again evaluated using the same traditional metrics.

The 2LE-BO-DeepTrade prediction framework proposed in this study differs from the ICE2DE-MDL model in three fundamental aspects: 1. Model selection: In contrast to the hybrid approach of ICE2DE-MDL, our framework focuses exclusively on deep learning methods. This aims to fully leverage the potential power of deep learning in capturing complex time-series patterns.

2. Hyperparameter optimization: Replacing the manual and potentially time-consuming hyperparameter tuning process of ICE2DE-MDL, the proposed framework employs an optimization algorithm to determine hyperparameters automatically and systematically. This potentially leads to improved model performance and enhances reproducibility.

3. Performance evaluation: In addition to traditional statistical error metrics, the evaluation incorporates a PLR-based trading strategy, developed within this study, to measure the practical financial applicability of our model. This aims to assess not only the statistical accuracy of the model but also its potential effectiveness in real-world trading scenarios.

The 2LE-BO-DeepTrade prediction framework

This study presents an integrated deep learning-based forecasting framework for predicting the next-day closing price of stocks. The proposed model1 combines the 2LE-ICEEMDAN denoising approach and optimized LSTM, LSTM-BN, and GRU methods. The model’s performance was analyzed not only with statistical evaluation metrics but also with the PLR-based trading strategy developed in this study.

The proposed 2LE-BO-DeepTrade framework for stock price prediction, as schematized in Fig. 3, consists of three main stages: 1. Denoising with 2LE-ICEEMDAN: The 2LE-ICEEMDAN denoising method is applied to the stock closing price data. The noiseless IMF components obtained in this method’s first and second decomposition stages, determined according to entropy values, are selected for model training.

2. Construction of the prediction model: Since each IMF component exhibits different dynamic characteristics, a separate modeling approach is adopted for each IMF instead of predicting all components with a single model. Accordingly, using Bayesian optimization, the optimal hyperparameters of LSTM, LSTM-BN, and GRU models are determined for each IMF. Each IMF is predicted separately with the three models trained with these optimized hyperparameters, and the model that provides the lowest MSE value is accepted as the final model for that IMF. The IMFs obtained from the first and second decomposition levels and predicted with the best models are summed to create the final prediction result.

3. Statistical and financial performance evaluation of the model: The model’s prediction performance is comprehensively evaluated and validated through both statistical metrics and the PLR-based trading strategy developed in this study.

Figure 3 The flowchart of the proposed 2LE-BO-DeepTrade.

Experimental settings

This study uses Python version 3.10 to implement the classification task for the identification of cerebral infarcts, the detailed configuration equipment is CPU: 12th Gen Intel(R) Core(TM) i9-12900K CPU @ 3.20 GHz, 16 cores, 24 logical processors, 64-bit Windows operating system.

This section first describes the stock datasets used in the study. Then, the search space used in the hyperparameter optimization of the deep learning models is detailed.

Dataset

To evaluate the performance of the proposed prediction framework, five stocks (AKBNK, MGROS, KCHOL, THYAO, and ULKER) from the BIST 100 index, which represents the 100 stocks with the highest market capitalization traded on Borsa Istanbul (BIST), were selected. These stocks were chosen because they are actively traded in financial markets, represent different sectors (banking, retail, holding, aviation, and food), and have high liquidity. Daily closing prices for the selected stocks between January 1, 2010, and January 1, 2020, were obtained from Yahoo Finance (https://finance.yahoo.com/), a reliable financial data source.

Following the 2LE-ICEEMDAN denoising process, each IMF for each stock was partitioned for model training and evaluation. Specifically, each IMF time series was divided into training and testing sets using an 80–20% split, respectively, maintaining chronological order. The first 80% of the data points were used for training (which includes the portion used for validation during Bayesian optimization), and the remaining 20% were reserved for testing the final model’s performance. Furthermore, prior to model input, both the training and testing data for each IMF were scaled to a range of [−1, 1] using the StandardScaler method.

Hyperparameter optimization

In the proposed model, hyperparameter optimization of the LSTM, LSTM-BN, and GRU methods was performed using Bayesian optimization. The optimization process aimed to find the best combination by evaluating the mean squared error performance of the model for each hyperparameter combination over 25 iterations. The search space determined in this process is detailed below: Number of layers: The search space for the number of layers used in the model was set as [1, 6]. Optimization of the number of layers is critically important for increasing the model’s capacity and learning ability. More layers allow the model to learn more complex structures, while selecting the appropriate number of layers reduces the risk of overfitting.

Dropout rate: The search space for this rate was set as [0.001, 0.5] to prevent the model from overfitting. Dropout improves the model’s generalization ability by randomly disabling certain neurons at each training step. The ideal dropout rate prevents the model from overfitting the training data, while enabling it to perform better on test data.

Learning rate: The search space for the learning rate parameter was set as [ 1×10−5, 1×10−2]. A low learning rate allows the model to learn more carefully by making smaller changes at each step, but a very low rate can slow down the model’s learning process. A high learning rate can enable the model to learn faster, but an excessively high rate can cause the model to diverge without reaching the optimal solution.

Batch size: This parameter refers to the number of samples used in each training step, and the search space for this parameter was set as [8, 256]. The optimal batch size both improves the model’s performance and makes memory usage more efficient.

Number of neurons in layers: The search space for the number of neurons in each layer was set as [8, 1,024]. The number of neurons directly affects each layer’s capacity and learning power. The ideal number of neurons allows the model to learn efficiently while avoiding unnecessary complexity.

Results and discussion

This section evaluated the 2LE-BO-DeepTrade, the proposed deep learning-based integrated stock prediction framework, through experimental studies conducted on different stocks. All stages of the approach developed within the scope of the study were examined and explained in detail on AKBNK, MGROS, KCHOL, THYAO, and ULKER stocks, and the results that were obtained were discussed.

In the first stage, the 2LE-ICEEMDAN denoising method was applied to the closing prices of the five stocks (AKBNK, MGROS, KCHOL, THYAO, and ULKER). For each stock, the entropy values and ratios of the IMFs obtained in the first and second decomposition stages are given in Table 3 ( Imf−i(j) means the i-th IMF as a result of the j-th decomposition). The values shown in bold in the table indicate the IMFs classified as high-frequency, in other words, noisy. The IMF components obtained from the decomposition are visualized in Figs. 4, and 5; here, for example, it is seen that nine IMFs were obtained in the first decomposition stage of the AKBNK stock, and two of them were classified as noisy (Fig. 4A). Similarly, nine IMFs were obtained in the second decomposition stage, and two were identified as noisy. The same analyses were repeated for the other four stocks (MGROS, KCHOL, THYAO, and ULKER), and the number of IMF components and noisy components for each stock were obtained similarly.

Table 3 Entropy values and ratios of IMFs acquired through the implementation of two decomposition stages for five stocks.

Stock	(i) Initial decomposition	(ii) Secondary decomposition	
	Approximate entropy	Sample entropy	Approximate entropy ratio	Sample entropy ratio		Approximate entropy	Sample entropy	Approximate entropy ratio	Sample entropy ratio	
AKBNK	Imf−1(′)	0.023	0.019	9.452	9.043	Imf−1(′′)	0.0012	0.00014	1.187	0.194	
Imf−2(′)	0.028	0.020	11.331	9.343	Imf − 2 (′′)	0.0534	0.03803	52.849	52.768	
Imf − 3 (′)	0.054	0.043	22.003	20.176	Imf − 3 (′′)	0.0367	0.02777	36.261	38.532	
Imf − 4 (′)	0.049	0.043	20.226	20.050	Imf−4(′′)	0.0038	0.00341	3.769	4.732	
Imf−5(′)	0.041	0.040	16.914	18.939	Imf−5(′′)	0.0013	0.00120	1.286	1.665	
Imf−6(′)	0.021	0.019	8.678	8.701	Imf−6(′′)	0.0011	0.00038	1.088	0.527	
Imf−7(′)	0.022	0.023	9.067	10.791	Imf−7(′′)	0.0012	0.00038	1.187	0.527	
Imf−8(′)	0.005	0.005	1.940	2.366	Imf−8(′′)	0.0014	0.00038	1.385	0.527	
	Imf−9(′)	0.001	0.001	0.389	0.590	Imf−9(′′)	0.0010	0.00038	0.989	0.527	
KCHOL	Imf − 1 (′)	0.118	0.084	21.6805	19.3668	Imf − 1 (′′)	0.08333	0.05738	98.6154	94.0656	
Imf−2(′)	0.062	0.050	11.3532	11.5015	Imf−2(′′)	0.00021	0.00039	0.2485	0.6393	
Imf−3(′)	0.106	0.076	19.5974	17.7039	Imf−3(′′)	0.00018	0.00038	0.2130	0.6230	
Imf−4(′)	0.082	0.067	15.0535	15.5777	Imf−4(′′)	0.00015	0.00038	0.1775	0.6230	
Imf−5(′)	0.086	0.066	15.7553	15.2720	Imf−5(′′)	0.00015	0.00035	0.1775	0.5738	
Imf−6(′)	0.050	0.046	9.1946	10.7210	Imf−6(′′)	0.00012	0.00034	0.1420	0.5574	
Imf−7(′)	0.026	0.027	4.7796	6.2302	Imf−7(′′)	0.00012	0.00035	0.1420	0.5738	
Imf−8(′)	0.012	0.013	2.2508	2.9900	Imf−8(′′)	0.00011	0.00037	0.1302	0.6066	
Imf−9(′)	0.002	0.003	0.3352	0.6369	Imf−9(′′)	0.00011	0.00036	0.1302	0.5902	
					Imf−10(′′)	0.00001	0.00038	0.0118	0.6230	
						Imf−11(′′)	0.00001	0.00032	0.0118	0.5246	
MGROS	Imf −1 (′)	0.451	0.294	23.3807	20.6336	Imf − 1 (′′)	0.40048	0.24381	89.9712	93.1568	
Imf−2(′)	0.297	0.188	15.3615	13.1507	Imf−2(′′)	0.04374	0.01485	9.8266	5.6740	
Imf−3(′)	0.366	0.260	18.9790	18.2550	Imf−3(′′)	0.00018	0.00057	0.0404	0.2178	
Imf−4(′)	0.293	0.236	15.1791	16.5178	Imf−4(′′)	0.00016	0.0004	0.0359	0.1528	
Imf−5(′)	0.216	0.183	11.1638	12.7953	Imf−5(′′)	0.00015	0.00038	0.0337	0.1452	
Imf−6(′)	0.137	0.123	7.0807	8.6571	Imf−6(′′)	0.00015	0.00038	0.0337	0.1452	
Imf−7(′)	0.111	0.089	5.7666	6.2574	Imf−7(′′)	0.00013	0.00036	0.0292	0.1376	
Imf−8(′)	0.051	0.046	2.6619	3.1918	Imf−8(′′)	0.00011	0.00034	0.0247	0.1299	
Imf−9(′)	0.008	0.007	0.3952	0.4753	Imf−9(′′)	0.00001	0.00033	0.0022	0.1261	
	Imf−10(′)	0.001	0.001	0.0316	0.0659	Imf−10(′′)	0.00001	0.0003	0.0022	0.1146	
THYAO	Imf − 1 (′)	0.144	0.066	24.5618	19.4417	Imf − 1 (′′)	0.0994	0.0413	83.5409	84.7607	
Imf−2(′)	0.075	0.040	12.8982	11.8763	Imf−2(′′)	0.0037	0.0025	3.0935	5.1756	
Imf−3(′)	0.095	0.047	16.2480	13.9004	Imf−3(′′)	0.0032	0.0014	2.7236	2.8343	
Imf−4(′)	0.108	0.058	18.4179	17.1757	Imf−4(′′)	0.0029	0.0010	2.4168	2.0949	
Imf−5(′)	0.066	0.048	11.2977	14.0539	Imf−5(′′)	0.0025	0.0006	2.0678	1.1912	
Imf−6(′)	0.043	0.031	7.3802	9.1381	Imf−6(′′)	0.0022	0.0005	1.8131	0.9448	
	Imf−7(′)	0.037	0.031	6.3935	9.0171	Imf−7(′′)	0.0018	0.0004	1.4764	0.8010	
Imf−8(′)	0.014	0.015	2.4658	4.4201	Imf−8(′′)	0.0013	0.0004	1.0565	0.7804	
Imf−9(′)	0.002	0.003	0.3369	0.9767	Imf−9(′′)	0.0011	0.0004	0.9541	0.7599	
						Imf−10(′′)	0.0010	0.0003	0.8574	0.6572	
ULKER	Imf − 1 (′)	0.291	0.179	24.7807	21.5557	Imf − 1 (′′)	0.2275	0.1317	85.6738	88.0313	
Imf−2(′)	0.209	0.141	17.8018	16.9717	Imf−2(′′)	0.0061	0.0055	2.2937	3.6688	
Imf−3(′)	0.218	0.158	18.5836	19.0634	Imf−3(′′)	0.0056	0.0042	2.1178	2.8268	
Imf−4(′)	0.174	0.122	14.8434	14.6966	Imf−4(′′)	0.0052	0.0031	1.9653	2.0984	
Imf−5(′)	0.110	0.087	9.3361	10.5058	Imf−5(′′)	0.0046	0.0022	1.7420	1.4635	
Imf−6(′)	0.099	0.081	8.3985	9.7687	Imf−6(′′)	0.0041	0.0008	1.5529	0.5213	
Imf−7(′)	0.045	0.035	3.8305	4.2559	Imf−7(′′)	0.0038	0.0006	1.4256	0.4143	
Imf−8(′)	0.019	0.017	1.6214	2.0664	Imf−8(′′)	0.0032	0.0004	1.1928	0.2874	
Imf−9(′)	0.009	0.009	0.7911	1.0338	Imf−9(′′)	0.0025	0.0004	0.9277	0.2606	
Imf−10(′)	0.0001	0.001	0.0128	0.0820	Imf−10(′′)	0.0017	0.0004	0.6346	0.2539	
					Imf−11(′′)	0.0013	0.0003	0.4738	0.1738	
Note:

The values shown in bold in the table indicate the IMFs classified as high-frequency, in other words, noisy.

Figure 4 First and second decomposition results of (A) AKBNK, (B) KCHOL, and (C) MGROS closing prices.

Figure 5 First and second decomposition results of (A) THYAO and (B) ULKER closing prices.

The noiseless IMF components obtained from the denoising stage were used in the construction of the prediction models. Accordingly, first, the optimal hyperparameters of LSTM, LSTM-BN, and GRU models were determined for each IMF using Bayesian optimization, and the model that yielded the lowest MSE value among the models trained with these hyperparameters was accepted as the final model for that IMF. The predictions of the IMFs predicted with these best models were summed to create the final prediction result. The hyperparameters obtained from this optimization process performed for the five stocks are presented in Tables 4–8. The final prediction results obtained in this way were used in the next stage to evaluate the model’s performance. In this context, first, the prediction results on the five stocks were evaluated using the statistical metrics presented in Table 9. Additionally, for comparison purposes, the prediction results of the ICE2DE-MDL model, which does not include hyperparameter optimization and was proposed in the study (Akşehir & Kılıç, 2024b), were also added to the same table. When the table is examined, it is observed that the proposed model, although having a simpler structure than the ICE2DE-MDL model, yields more successful results thanks to the best method architectures determined by hyperparameter optimization. This finding demonstrates that model complexity does not always lead to better results; on the contrary, simpler structures can achieve similar or better performance. Furthermore, graphs visualizing the prediction results for each dataset are presented in Figs. 6 and 7 for a detailed examination of the model performance. In the figures, the green curve represents the actual closing values of the stocks, the blue curve represents the data obtained after the denoising process, the red curve represents the predictions of the ICE2DE-MDL model, and the orange curve represents the predictions of our proposed model. When the graphs are examined, it is seen that the predictions of the ICE2DE-MDL model, which used denoised data but did not undergo hyperparameter optimization, have lower accuracy than our proposed model’s predictions. These results demonstrate that supporting our proposed model with hyperparameter optimization improves prediction accuracy by providing a more suitable configuration and parameter selection, thus contributing to obtaining more reliable results.

Table 4 Hyperparameter optimization results for AKBNK stock.

IMF	Deep learning model	Number of layers	Dropout rate	Learning rate	Batch size	Number of neurons	
Imf−1(′)	LSTM	1	0.4764	0.00543	8	[8]	
GRU	1	0.0213	0.00447	8	[9]	
	LSTM-BN	1	0.2756	0.00184	8	[31]	
Imf−2(′)	LSTM	1	0.4330	0.00211	8	[570]	
GRU	4	0.1841	0.00105	53	[296, 841, 506, 22]	
	LSTM-BN	1	0.2358	0.00429	8	[8]	
Imf−3(′)	LSTM	2	0.0010	0.00159	8	[783, 1,024]	
GRU	1	0.5000	0.00424	89	[137]	
	LSTM-BN	1	0.2899	0.00268	8	[8]	
Imf−4(′)	LSTM	1	0.2755	0.00344	58	[451]	
GRU	1	0.3832	0.01000	8	[323]	
	LSTM-BN	1	0.5000	0.00547	8	[8]	
Imf−5(′)	LSTM	1	0.4213	0.00648	217	[277]	
GRU	1	0.4213	0.00648	217	[277]	
	LSTM-BN	6	0.4616	0.00133	8	[949, 954, 195, 79, 1,000, 8]	
Imf−6(′)	LSTM	1	0.2722	0.00821	8	[8]	
GRU	1	0.0010	0.00357	8	[8]	
	LSTM-BN	1	0.2926	0.00217	8	[8]	
Imf−7(′)	LSTM	1	0.4252	0.00471	102	[94]	
GRU	1	0.1753	0.00437	248	[593]	
	LSTM-BN	1	0.0010	0.00571	8	[8]	
Imf−1(′′)	LSTM	1	0.5000	0.01000	8	[8]	
GRU	1	0.0845	0.00737	8	[460]	
	LSTM-BN	1	0.4213	0.00648	217	[277]	
Imf−2(′′)	LSTM	1	0.5000	0.00558	8	[8]	
GRU	1	0.4213	0.00648	217	[277]	
	LSTM-BN	1	0.0010	0.00876	8	[8]	
Imf−3(′′)	LSTM	1	0.0010	0.00498	33	[415]	
GRU	1	0.0010	0.00516	19	[770]	
	LSTM-BN	1	0.3718	0.00057	8	[144]	
Imf−4(′′)	LSTM	1	0.5000	0.01000	8	[8]	
GRU	1	0.4213	0.00648	217	[277]	
	LSTM-BN	1	0.3416	0.00312	8	[375]	
Imf−5(′′)	LSTM	1	0.0010	0.01000	8	[8]	
GRU	1	0.2638	0.00642	8	[8]	
	LSTM-BN	1	0.2296	0.00001	8	[948]	
Imf−6(′′)	LSTM	1	0.3651	0.00670	114	[143]	
GRU	1	0.4213	0.00648	217	[277]	
	LSTM-BN	2	0.0732	0.00612	8	[810, 117]	
Imf−7(′′)	LSTM	1	0.4213	0.00648	217	[277]	
GRU	6	0.5000	0.00001	8	[1,024, 1,024, 1,024, 1,024, 8, 486]	
	LSTM-BN	1	0.0010	0.00381	8	[311]	

Table 5 Hyperparameter optimization results for KCHOL stock.

IMF	Deep learning model	Number of layers	Dropout rate	Learning rate	Batch size	Number of neurons	
Imf−1(′)	LSTM	1	0.3175	0.0031	15	[221]	
GRU	1	0.0906	0.0034	8	[66]	
	LSTM-BN	1	0.4493	0.0022	8	[154]	
Imf−2(′)	LSTM	1	0.5000	0.0040	8	[77]	
GRU	1	0.2170	0.0080	78	[202]	
	LSTM-BN	1	0.1263	0.0017	11	[1,014]	
Imf−3(′)	LSTM	1	0.1641	0.0044	8	[382]	
GRU	1	0.2381	0.0017	8	[368]	
	LSTM-BN	1	0.5000	0.0100	8	[8]	
Imf−4(′)	LSTM	1	0.0244	0.0047	16	[313]	
GRU	1	0.0456	0.0031	8	[8]	
	LSTM-BN	1	0.4213	0.0065	217	[277]	
Imf−5(′)	LSTM	1	0.4213	0.0065	217	[277]	
GRU	1	0.4621	0.0028	89	[694]	
	LSTM-BN	3	0.3377	0.0032	201	[973, 681, 22]	
Imf−6(′)	LSTM	1	0.4851	0.0068	212	[161]	
GRU	1	0.0624	0.0028	8	[109]	
	LSTM-BN	1	0.0616	0.0014	8	[1,024]	
Imf−7(′)	LSTM	1	0.4213	0.0065	217	[277]	
GRU	1	0.4737	0.0077	8	[8]	
	LSTM-BN	1	0.3565	0.0027	19	[8]	
Imf−8(′)	LSTM	1	0.4213	0.0065	217	[277]	
GRU	3	0.0010	0.0026	8	[8, 41, 467]	
	LSTM-BN	5	0.2548	0.0022	8	[513, 8, 733, 400, 520]	
Imf−1(′′)	LSTM	1	0.3241	0.0013	8	[223]	
GRU	1	0.2550	0.0023	8	[735]	
	LSTM-BN	1	0.4213	0.0065	217	[277]	
Imf−2(′′)	LSTM	1	0.5000	0.0044	10	[8]	
GRU	1	0.2491	0.0028	8	[8]	
	LSTM-BN	1	0.3670	0.0025	53	[359]	
Imf−3(′′)	LSTM	1	0.5000	0.0056	8	[236]	
GRU	1	0.5000	0.0033	8	[329]	
	LSTM-BN	2	0.3687	0.0022	42	[337, 160]	
Imf−4(′′)	LSTM	1	0.5000	0.0038	8	[464]	
GRU	2	0.1807	0.0058	48	[164, 290]	
	LSTM-BN	1	0.0010	0.0100	8	[8]	
Imf−5(′′)	LSTM	1	0.0358	0.0084	50	[79]	
GRU	1	0.1086	0.0058	40	[192]	
	LSTM-BN	1	0.4325	0.0040	8	[109]	
Imf−6(′′)	LSTM	1	0.0010	0.0098	8	[8]	
GRU	1	0.4213	0.0065	217	[277]	
	LSTM-BN	1	0.4761	0.0100	8	[8]	
Imf−7(′′)	LSTM	1	0.4213	0.0065	217	[277]	
GRU	1	0.0010	0.0045	60	[352]	
	LSTM-BN	1	0.0010	0.0021	8	[8]	
Imf−8(′′)	LSTM	2	0.3687	0.0022	42	[337, 160]	
GRU	1	0.4213	0.0065	217	[277]	
	LSTM-BN	2	0.3687	0.0022	42	[337, 160]	
Imf−9(′′)	LSTM	1	0.4213	0.0065	217	[277]	
GRU	1	0.4213	0.0065	217	[277]	
	LSTM-BN	1	0.4388	0.0016	8	[844]	
Imf−10(′′)	LSTM	1	0.0779	0.0014	42	[608]	
GRU	2	0.3687	0.0022	42	[337, 160]	
	LSTM-BN	3	0.3377	0.0032	201	[973, 681, 22]	

Table 6 Hyperparameter optimization results for MGROS stock.

IMF	Deep learning model	Number of layers	Dropout rate	Learning rate	Batch size	Number of neurons	
Imf−1(′)	LSTM	2	0.3954	0.0022	13	[249, 140]	
GRU	2	0.3687	0.0022	42	[337, 160]	
	LSTM-BN	1	0.5000	0.0100	234	[8]	
Imf−2(′)	LSTM	1	0.0659	0.0097	8	[54]	
GRU	1	0.4093	0.0039	104	[171]	
	LSTM-BN	1	0.4254	0.0058	71	[87]	
Imf−3(′)	LSTM	1	0.0859	0.0024	13	[297]	
GRU	1	0.0246	0.0017	24	[417]	
	LSTM-BN	2	0.3687	0.0022	42	[337, 160]	
Imf−4(′)	LSTM	1	0.0710	0.0100	21	[8]	
GRU	1	0.5000	0.0031	17	[258]	
	LSTM-BN	2	0.3687	0.0022	42	[337, 160]	
Imf−5(′)	LSTM	5	0.0534	0.0015	28	[692, 271, 218, 35, 1,024]	
GRU	2	0.3687	0.0022	42	[337, 160]	
	LSTM-BN	2	0.3687	0.0022	42	[337, 160]	
Imf−6(′)	LSTM	1	0.3829	0.0012	100	[718]	
GRU	1	0.4213	0.0065	217	[277]	
	LSTM-BN	1	0.0010	0.0047	8	[1,024]	
Imf−7(′)	LSTM	1	0.4213	0.0065	217	[277]	
GRU	1	0.4213	0.0065	217	[277]	
	LSTM-BN	1	0.3025	0.0025	8	[8]	
Imf−8(′)	LSTM	1	0.4213	0.0065	217	[277]	
GRU	1	0.4213	0.0065	217	[277]	
	LSTM-BN	5	0.3834	0.0005	8	[1,007, 711, 1012, 661, 97]	
Imf−9(′)	LSTM	2	0.0010	0.00001	8	[1,024, 1,024]	
GRU	1	0.0010	0.0046	8	[259]	
	LSTM-BN	4	0.3514	0.0045	135	[917, 518, 337, 912]	
Imf−1(′′)	LSTM	1	0.4541	0.0028	8	[435]	
GRU	2	0.1800	0.0083	8	[89, 816]	
	LSTM-BN	1	0.3960	0.0007	230	[948]	
Imf−2(′′)	LSTM	1	0.3025	0.0100	8	[210]	
GRU	1	0.0831	0.0034	8	[317]	
	LSTM-BN	1	0.3899	0.0045	55	[184]	
Imf−3(′′)	LSTM	1	0.5000	0.0056	8	[218]	
GRU	1	0.4213	0.0065	217	[277]	
	LSTM-BN	1	0.5000	0.0100	8	[8]	
Imf−4(′′)	LSTM	1	0.4213	0.0065	217	[277]	
GRU	1	0.0177	0.0028	230	[576]	
	LSTM-BN	1	0.0010	0.0100	8	[8]	
Imf−5(′′)	LSTM	1	0.3126	0.0044	114	[446]	
GRU	1	0.3344	0.0026	8	[175]	
	LSTM-BN	1	0.4199	0.0057	213	[171]	
Imf−6(′′)	LSTM	5	0.0276	0.0015	8	[918, 548, 137, 841, 991]	
GRU	1	0.4213	0.0065	217	[277]	
	LSTM-BN	2	0.3994	0.0013	17	[368, 176]	
Imf−7(′′)	LSTM	1	0.4408	0.0058	130	[97]	
GRU	1	0.4213	0.0065	217	[277]	
	LSTM-BN	1	0.0010	0.0100	8	[8]	
Imf−8(′′)	LSTM	1	0.4213	0.0065	217	[277]	
GRU	1	0.4213	0.0065	217	[277]	
	LSTM-BN	1	0.0010	0.0079	8	[8]	
Imf−9(′′)	LSTM	2	0.5000	0.0002	48	[972, 279]	
GRU	1	0.1270	0.0030	69	[269]	
	LSTM-BN	2	0.3927	0.0019	8	[979, 688]	

Table 7 Hyperparameter optimization results for THYAO stock.

IMF	Deep learning model	Number of layers	Dropout rate	Learning rate	Batch size	Number of neurons	
Imf−1(′)	LSTM	1	0.3849	0.00233	8	[119]	
GRU	1	0.2412	0.00630	8	[132]	
	LSTM-BN	1	0.0010	0.01000	8	[8]	
Imf−2(′)	LSTM	1	0.3044	0.00352	9	[341]	
GRU	1	0.5000	0.01000	8	[8]	
	LSTM-BN	2	0.3687	0.00217	42	[337, 160]	
Imf−3(′)	LSTM	1	0.5000	0.00792	8	[279]	
GRU	3	0.0010	0.00285	8	[1,024, 8, 594]	
	LSTM-BN	1	0.4202	0.00635	214	[275]	
Imf−4(′)	LSTM	1	0.5000	0.00346	8	[1,024]	
GRU	1	0.4213	0.00648	217	[277]	
	LSTM-BN	1	0.1448	0.00964	94	[8]	
Imf−5(′)	LSTM	1	0.0010	0.00196	256	[857]	
GRU	2	0.0091	0.00724	8	[31, 8]	
	LSTM-BN	1	0.3854	0.00001	8	[1,024]	
Imf−6(′)	LSTM	1	0.3846	0.00203	21	[793]	
GRU	1	0.4213	0.00648	217	[277]	
	LSTM-BN	1	0.3913	0.00534	8	[26]	
Imf−7(′)	LSTM	1	0.0010	0.00158	32	[612]	
GRU	1	0.3192	0.00265	133	[161]	
	LSTM-BN	1	0.5000	0.00001	8	[764]	
Imf−8(′)	LSTM	1	0.2059	0.00034	183	[1,024]	
GRU	4	0.4513	0.00100	249	[672, 182, 372, 771]	
	LSTM-BN	1	0.5000	0.00582	8	[8]	
Imf−1(′′)	LSTM	3	0.4007	0.00521	176	[740, 599, 554]	
GRU	1	0.2996	0.00339	10	[66]	
	LSTM-BN	2	0.3721	0.00648	8	[8, 8]	
Imf−2(′′)	LSTM	2	0.0010	0.00456	8	[668, 8]	
GRU	1	0.4918	0.01000	8	[8]	
	LSTM-BN	1	0.5000	0.00333	8	[8]	
Imf−3(′′)	LSTM	2	0.5000	0.00050	8	[353, 202]	
GRU	1	0.0386	0.00727	51	[119]	
	LSTM-BN	1	0.3293	0.00710	8	[8]	
Imf−4(′′)	LSTM	1	0.5000	0.00173	8	[1,024]	
GRU	1	0.4269	0.00484	8	[118]	
	LSTM-BN	1	0.0010	0.00205	8	[8]	
Imf−5(′′)	LSTM	1	0.0430	0.00250	8	[72]	
GRU	1	0.1336	0.00611	58	[382]	
	LSTM-BN	2	0.4549	0.00707	8	[632, 546]	
Imf−6(′′)	LSTM	1	0.0267	0.00129	8	[813]	
GRU	1	0.4384	0.00394	8	[588]	
	LSTM-BN	1	0.5000	0.00001	8	[1,024]	
Imf−7(′′)	LSTM	1	0.4213	0.00648	217	[277]	
GRU	1	0.4213	0.00648	217	[277]	
	LSTM-BN	1	0.2265	0.00118	8	[8]	
Imf−8(′′)	LSTM	1	0.4213	0.00648	217	[277]	
GRU	1	0.4213	0.00648	217	[277]	
	LSTM-BN	1	0.5000	0.01000	8	[8]	
Imf−9(′′)	LSTM	2	0.0010	0.00176	16	[8, 310]	
GRU	4	0.2937	0.00001	155	[866, 551, 8, 8]	
	LSTM-BN	6	0.5000	0.00013	219	[478, 669, 182, 1,024, 13, 1,024]	

Table 8 Hyperparameter optimization results for ULKER stock.

IMF	Deep learning model	Number of layers	Dropout rate	Learning rate	Batch size	Number of neurons	
Imf−1(′)	LSTM	1	0.0723	0.0075	8	[430]	
GRU	1	0.1441	0.0065	8	[362]	
	LSTM-BN	1	0.5000	0.0100	8	[32]	
Imf−2(′)	LSTM	1	0.0010	0.0100	8	[8]	
GRU	5	0.0010	0.0028	11	[8, 510, 8, 357, 66]	
	LSTM-BN	1	0.5000	0.0018	8	[375]	
Imf−3(′)	LSTM	1	0.4077	0.0072	47	[174]	
GRU	1	0.4213	0.0065	217	[277]	
	LSTM-BN	1	0.5000	0.0039	8	[8]	
Imf−4(′)	LSTM	1	0.5000	0.0079	152	[159]	
GRU	2	0.3687	0.0022	42	[337, 160]	
	LSTM-BN	1	0.0010	0.00001	8	[978]	
Imf−5(′)	LSTM	1	0.4634	0.0014	10	[1,001]	
GRU	1	0.0010	0.0052	14	[8]	
	LSTM-BN	2	0.3687	0.0022	42	[337, 160]	
Imf−6(′)	LSTM	1	0.4196	0.0040	174	[421]	
GRU	1	0.4213	0.0065	217	[277]	
	LSTM-BN	2	0.4090	0.0020	46	[339, 177]	
Imf−7(′)	LSTM	1	0.0010	0.0044	8	[187]	
GRU	1	0.1057	0.0068	40	[224]	
	LSTM-BN	1	0.3885	0.0037	8	[8]	
Imf−8(′)	LSTM	2	0.3687	0.0022	42	[337, 160]	
GRU	1	0.1258	0.0048	149	[241]	
	LSTM-BN	1	0.1631	0.00002	8	[962]	
Imf−9(′)	LSTM	1	0.4213	0.0065	217	[277]	
GRU	1	0.3827	0.0006	190	[661]	
	LSTM-BN	3	0.3091	0.0023	195	[1,008, 648, 78]	
Imf−1(′′)	LSTM	1	0.4527	0.0040	11	[77]	
GRU	2	0.3687	0.0022	42	[337, 160]	
	LSTM-BN	1	0.0481	0.0100	8	[229]	
Imf−2(′′)	LSTM	1	0.1378	0.0051	9	[146]	
GRU	1	0.4062	0.0043	8	[635]	
	LSTM-BN	1	0.5000	0.0100	8	[8]	
Imf−3(′′)	LSTM	2	0.0981	0.0041	8	[80, 868]	
GRU	2	0.1738	0.0050	8	[158, 85]	
	LSTM-BN	5	0.1318	0.0020	8	[8, 12, 1,024, 72, 146]	
Imf−4(′′)	LSTM	1	0.2201	0.0060	8	[692]	
GRU	1	0.5000	0.0047	37	[364]	
	LSTM-BN	6	0.0010	0.00001	8	[8, 513, 566, 8, 8, 1,024]	
Imf−5(′′)	LSTM	2	0.0010	0.0073	8	[8, 29]	
GRU	1	0.0010	0.0043	8	[1,024]	
	LSTM-BN	1	0.0010	0.00001	8	[648]	
Imf−6(′′)	LSTM	1	0.5000	0.0019	8	[692]	
GRU	1	0.4213	0.0065	217	[277]	
	LSTM-BN	1	0.0010	0.0001	8	[8]	
Imf−7(′′)	LSTM	1	0.4213	0.0065	217	[277]	
GRU	1	0.3501	0.0068	41	[140]	
	LSTM-BN	1	0.5000	0.00001	8	[658]	
Imf−8(′′)	LSTM	1	0.4213	0.0065	217	[277]	
GRU	2	0.0024	0.0016	36	[637, 201]	
	LSTM-BN	2	0.3687	0.0022	42	[337, 160]	
Imf−9(′′)	LSTM	5	0.4965	0.0038	62	[991, 517, 807, 581, 594]	
GRU	1	0.5000	0.0012	113	[1,024]	
	LSTM-BN	1	0.0010	0.00001	8	[1,024]	
Imf−10(′′)	LSTM	2	0.3687	0.0022	42	[337, 160]	
GRU	1	0.5000	0.0092	8	[276]	
	LSTM-BN	5	0.0703	0.0056	24	[340, 195, 196, 8, 115]	

Table 9 Forecast results of the 2LE-BO-DeepTrade model for stocks.

Dataset	Statistical metrics	ICE2DE-MDL	2LE-BO-DeepTrade	
AKBNK	RMSE	0.177	0.014	
MAE	0.131	0.011	
MAPE	0.486	0.492	
	R 2	0.929	0.930	
MGROS	RMSE	0.070	0.004	
MAE	0.053	0.003	
MAPE	2.465	0.488	
	R 2	0.990	0.993	
KCHOL	RMSE	0.184	0.007	
MAE	0.114	0.006	
MAPE	0.315	0.202	
	R 2	0.976	0.992	
THYAO	RMSE	0.291	0.014	
MAE	0.183	0.010	
MAPE	0.656	0.317	
	R 2	0.957	0.985	
ULKER	RMSE	0.104	0.006	
MAE	0.071	0.005	
MAPE	0.225	0.179	
	R 2	0.988	0.994	

Figure 6 Graphics depicting the proposed prediction model’s performance on the (A) AKBNK, (B) KCHOL, and (C) MGROS stock test dataset.

Figure 7 Graphics depicting the proposed prediction model’s performance on the (A) THYAO and (B) ULKER stock test dataset.

The model’s performance was also evaluated financially using a PLR-based trading strategy. For this purpose, the strategy, with the number of segments set to 10, generated “Buy” and “Sell” signals for the stocks using the model’s predictions. Assuming an initial capital of 10,000 TL and applying a transaction fee of 0.1% for each trade, trading simulations were performed for the five stocks in the BIST 100 index. The buy and sell signals generated by this PLR-based strategy are shown in Figs. 8 and 9 (green: “Buy”, red: “Sell”), and the profit amounts obtained at the end of an approximately two-and-a-half-year period are shown in Table 10. It was observed that trading based on predictions yielded significantly higher returns compared to actual prices for THYAO, MGROS, and KCHOL stocks, but resulted in lower returns for AKBNK and ULKER stocks. This lower performance, especially for AKBNK, is due to the high volatility of the stock during the period in question. When compared to the “Buy and Hold Strategy” column in Table 10—representing a simple passive investment strategy where the stock is held for the entire period—the PLR-based strategy is seen to provide significantly higher returns for all stocks than this basic strategy. For example, 51 times more profit was obtained for AKBNK, 87 times for MGROS, 19 times for KCHOL, 15 times for THYAO, and 160 times for ULKER. These striking results demonstrate that the PLR-based trading strategy offers significantly higher return potential compared to passive investment strategies and that the model’s predictions can be effectively used to make profitable trading decisions.

Figure 8 Buy-sell signals for (A) AKBNK, (B) KCHOL and (C) MGROS stocks.

Figure 9 Buy-sell signals for (A) THYAO, and (B) ULKER stocks.

Table 10 Stock profitability analysis.

Stock	Proposed strategy	Buy and hold strategy	
AKBNK	17,640.59	−346.74	
MGROS	104,133.44	−1,192.87	
KCHOL	48,514.21	2,561.08	
THYAO	120,443.32	7,952.73	
ULKER	53,042.35	332.34	

Conclusion

In this study, an integrated deep learning-based forecasting framework, named 2LE-BO-DeepTrade, was presented to predict the next-day closing prices of stocks. The framework was evaluated through comprehensive experimental studies on five stocks from different sectors traded on Borsa Istanbul: AKBNK (banking), MGROS (retail), KCHOL (holding), THYAO (aviation), and ULKER (food). The main contributions of this study are: (1) To present a novel forecasting framework that uses the 2LE-ICEEMDAN method for denoising stock prices and combines it with optimized deep learning models. (2) To develop a practical and profitable PLR-based trading strategy based on this framework. The framework’s effectiveness was demonstrated through a multi-stage process involving denoising with the 2LE-ICEEMDAN method, hyperparameter optimization with Bayesian optimization, and performance evaluation using various statistical and financial metrics (trading strategy). The obtained results demonstrate that the proposed model, despite having a simpler structure compared to existing methods like ICE2DE-MDL, achieves superior prediction accuracy thanks to its optimized model architecture and hyperparameters. This finding indicates that complex models may not always yield better results; on the contrary, simpler structures that are appropriately optimized can achieve similar or even better performance. Visual and statistical comparisons between the models support the effectiveness of the proposed approach and show that the optimized hyperparameters significantly improve prediction accuracy and provide more reliable predictions. Furthermore, the PLR-based trading strategy developed in this study has contributed to making more effective investment decisions. Trading operations conducted in accordance with the “Buy” and “Sell” signals generated by the model with the PLR-based trading strategy have yielded higher profits compared to traditional strategies. The proposed strategy provided significantly higher returns for all stocks compared to the passive trading strategy. These results highlight the effectiveness of the proposed PLR-based trading strategy and the potential returns it can provide to investors.

In conclusion, the 2LE-BO-DeepTrade forecasting framework proposed in this study enables accurate and reliable stock price prediction and allows profitable investment decisions to be made using these predictions. This framework, which integrates denoising, optimized deep learning modeling, and financial evaluation, offers a new approach to the field of stock price prediction.

Limitations and future work

While this study demonstrates a promising approach for stock price prediction and trading, it is important to acknowledge certain limitations. The analysis is constrained by the specific timeframe and the limited set of stocks analyzed, and the model’s performance may vary under different market conditions or with different stocks. Furthermore, the PLR-based trading strategy, despite showing significant profitability, also has its limitations. The strategy’s performance, particularly in highly volatile market conditions, was not specifically investigated in this study, and its sensitivity to the number of segments (N), a critical parameter, was not analyzed. Although it is anticipated that the profitability of the PLR strategy might decrease in market conditions involving high volatility, we believe that this profitability can be maximized with an N parameter that is dynamically selected or optimized according to market conditions.

Future research should focus on enhancing the model’s generalizability by expanding the dataset and exploring diverse market conditions. Specifically for the PLR strategy, a comprehensive sensitivity analysis for the ‘N’ parameter should be conducted in line with the consideration above. Methods to optimize this parameter for different market regimes (especially during periods of high volatility) should be investigated, and the strategy’s resilience and effectiveness under various market volatility scenarios should be evaluated accordingly. In addition, exploring alternative hyperparameter optimization techniques, different trading strategies, and incorporating external factors such as macroeconomic indicators or market sentiment could further improve the model’s overall accuracy and robustness. Finally, future work should validate the model’s performance in real-world trading environments and investigate advanced techniques to capture complex market dynamics.

Supplemental Information

Supplemental Information 1 AKBNK stock dataset.

Supplemental Information 2 THYAO stock dataset.

Supplemental Information 3 MGROS stock dataset.

Supplemental Information 4 KCHOL stock dataset.

Supplemental Information 5 ULKER stock dataset.

Supplemental Information 6 Reproducibility.

Supplemental Information 7 README.

Additional Information and Declarations

Competing Interests

The authors declare that they have no competing interests.

Author Contributions

Zinnet Duygu Akşehir conceived and designed the experiments, performed the experiments, analyzed the data, performed the computation work, prepared figures and/or tables, authored or reviewed drafts of the article, and approved the final draft.

Erdal Kılıç conceived and designed the experiments, performed the experiments, analyzed the data, authored or reviewed drafts of the article, and approved the final draft.

Data Availability

The following information was supplied regarding data availability:

The data and code is available at GitHub and Zenodo:

- https://github.com/aksehird/2LE-BO-DeepTrade,

- KILIÇ, E., & AKŞEHİR, Z. D. (2025). 2LE-BO-DeepTrade. https://doi.org/10.5281/zenodo.15187124.

1 The source code related to the proposed model is available at Zenodo (DOI: 10.5281/zenodo.15187124) and on GitHub: https://github.com/aksehird/2LE-BO-DeepTrade.

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
