# Peer review of "LE-BO-DeepTrade: an integrated deep learning framework for stock price prediction"

_PeerJ Computer Science, doi:10.7717/peerj-cs.3107_

## Round 0.1 · original submission · Major Revisions

**Language Note:** The review process has identified that the English language must be improved. PeerJ can provide language editing services - please contact us at [email protected] for pricing (be sure to provide your manuscript number and title). Alternatively, you should make your own arrangements to improve the language quality and provide details in your response letter. – PeerJ Staff

·

Basic reporting

What is the role of the 2LE-ICEEMDAN denoising method in improving time-series prediction in the 2LE-BO-DeepTrade framework?

How do LSTM, LSTM-BN, and GRU differ in handling temporal dependencies in stock market data?

What advantages does Bayesian Optimization offer in tuning deep learning models compared to grid or random search in this framework?

Explain how the Piecewise Linear Representation (PLR) contributes to the design of a trading strategy in the model.

Why might denoising be particularly important in financial time series forecasting?

What is the purpose of decomposing stock data into Intrinsic Mode Functions (IMFs)?

In what way does the framework determine the optimal model for each IMF component?

What makes LSTM with Batch Normalization (LSTM-BN) potentially superior to standard LSTM in financial prediction tasks?

How does the integrated approach of model optimization and trading logic distinguish this study from previous models like ICE2DE-MDL?

Why might a multi-model ensemble strategy be more effective than a single model in stock price prediction?

Experimental design

How can this framework be adapted to forecast stock prices in markets outside of Borsa Istanbul?

What are the practical implications of the reported 160x gain when using the PLR-based strategy compared to passive investing?

In a real-time application, how would this framework handle non-stationary or abrupt market events like crashes or political shocks?

How scalable is the proposed framework to larger datasets or high-frequency trading data?

What kind of investors (e.g., retail, institutional, algorithmic) would most benefit from using the 2LE-BO-DeepTrade strategy?

How would transaction costs, slippage, and market liquidity affect the practical returns generated by the PLR-based strategy?

Could this model be extended to include multivariate time series inputs like macroeconomic indicators or sentiment scores?

How could regulators or market analysts benefit from prediction models like 2LE-BO-DeepTrade?

What are the deployment considerations (e.g., computational cost, latency) in applying this model in real-world financial systems?

How might the system be integrated with automated trading platforms such as MetaTrader, QuantConnect, or Alpaca?

Validity of the findings

How does the model's accuracy compare against traditional statistical models like ARIMA or Prophet?

Explain how the framework measures success using both statistical metrics (like RMSE or MAE) and financial metrics (like ROI).

Why is it important to evaluate predictive performance on multiple stocks from different sectors?

What are the strengths and weaknesses of comparing 2LE-BO-DeepTrade with a single benchmark like ICE2DE-MDL?

How robust is the model's performance under cross-validation or walk-forward validation techniques?

How might the results vary if other noise reduction techniques like wavelet transforms or SSA were used instead of 2LE-ICEEMDAN?

Why might the framework prioritize interpretability and simplicity over more complex models in financial applications?

How well does the model generalize to out-of-sample predictions or unseen financial instruments?

What risks exist in overfitting when tuning models with Bayesian Optimization in this context?

Can the PLR-based strategy be objectively validated using backtesting frameworks like QuantLib or Zipline?

Additional comments

This paper, while presenting an integrated deep learning framework for stock price prediction, lacks sufficient novelty and methodological transparency to warrant publication. The approach primarily combines existing techniques—such as ICEEMDAN denoising, LSTM/GRU models, and Bayesian optimization—without offering substantial theoretical advancement or innovation beyond previous studies.

Furthermore, critical details regarding the design of the PLR-based trading strategy, evaluation protocols, and statistical validation are either underexplained or missing, limiting the reproducibility and generalizability of the results. The impressive claims of returns (e.g., 160x profits) lack adequate empirical justification and risk modeling, raising concerns about overfitting or selective reporting. Overall, the manuscript falls short in demonstrating a clear, novel contribution to the field of financial forecasting using deep learning.

·

Basic reporting

The abstract provides a comprehensive overview of the proposed 2LE-BO-DeepTrade framework and its methodological innovations. However, it lacks specific quantitative results regarding the model's predictive performance. Including key statistical metrics (e.g., RMSE, MAE, MAPE, or R²) or comparative performance outcomes in the abstract would strengthen the presentation and help readers quickly grasp the effectiveness of the proposed approach

The Introduction section requires a complete rewrite to better align with the objectives of the study. Currently, the introduction does not clearly define the problem statement or provide a logical flow that leads to the proposed solution. Additionally, the content is not directly relevant to the core contributions of your work. For instance, in line 57, model evaluation is discussed prematurely—this belongs in the "Results and Discussion" section. The Introduction should instead focus on the motivation behind the study, existing gaps in the literature, and a clear articulation of how your proposed framework addresses those gaps. Please revise accordingly to improve clarity, coherence, and structure.

In line 239, the manuscript states, “For the first time in the literature, an integrated approach is presented…” However, similar approaches that combine denoising techniques, Bayesian optimization, and deep learning models for stock price prediction can be found in the existing literature. Therefore, this claim appears to be overstated. Please either provide a strong justification with proper citations to support the "first time" claim or revise the sentence to avoid making an unsubstantiated novelty claim..
https://www.sciencedirect.com/science/article/abs/pii/S0020025522000135
https://link.springer.com/article/10.1007/s10614-023-10393-4

Experimental design

Please include a table summarizing the related work along with their merits and demerits, and explain how each is related to your problem statement.

In line no 467, Daily closing prices for the selected stocks between January 1, 467 2010, and January 1, 2020,. Why 2020 only? I recommend that you kindly consider including at least one recent dataset from the year 2024 to enhance the relevance and applicability of your study



A more detailed explanation of the denoising process is needed in the Methodology section, including the steps involved, key parameters used, how these parameters were selected or tuned.
Please review Figure 1, as it appears to be directly taken from existing literature. This raises potential plagiarism concerns. I strongly recommend that you redesign Figure 1 and Figure2.

Validity of the findings

In line 469, it is stated that hyperparameter optimization of the LSTM, LSTM-BN, and GRU models was performed using Bayesian optimization. Please justify why Bayesian optimization was selected. Additionally, clearly define the objective function used in the optimization process.
In Table 1, please provide a detailed description of the variables Pi and Ai in the main text.

The results are not compared with the existing state-of-the-art methods reported in the literature. Please include a comparative analysis to justify the effectiveness of your proposed model relative to other recent approaches.
Add this citation, PeerJ Computer Science
Novel grey wolf optimizer based parameters selection for GARCH and ARIMA models for stock price prediction

Additional comments

Major Revision

·

Basic reporting

1. Abstract should be concise, describing the objective, methodology and the key outcome. Avoid the term, "for example, in conclusion etc."
2.There are recurring minor grammatical and typographical issues (e.g., "Funsitons" instead of "Functions", line 20).
3. Certain concepts (e.g., justification of PLR, benefits of denoising) are repeatedly explained across multiple sections
4. The integration of denoising, Bayesian optimization, and PLR-based trading is a strength, although similar hybrid approaches exist. The novelty is incremental rather than groundbreaking.

Experimental design

1. The search space for Bayesian optimization is not well-justified. Why were those specific ranges chosen (e.g., neurons between 8 and 1024)?
2. The number of segments in the PLR-based trading strategy is arbitrarily set to 10, with no sensitivity analysis or justification for this choice. The effectiveness of PLR may vary significantly with this parameter.
3. Bayesian optimization was performed using a small dataset (single stock over 10 years). There is no mention of cross-validation, regularization, or other techniques to guard against overfitting.

Validity of the findings

1. Claims like “160 times higher return than passive strategy” (line 23) are exaggerated without risk-adjusted comparison or reference to drawdown/volatility metrics.
2. While PLR is promising, its limitations in volatile markets or its sensitivity to segment count are not acknowledged or discussed.
3. With hyperparameter optimization and small datasets, overfitting risk is high, but no mitigation (e.g., cross-validation) is discussed.
4. The manuscript reports performance improvements but provides no statistical testing (e.g., p-values, confidence intervals, t-tests) to demonstrate whether the improvements are statistically meaningful.

Additional comments

1. The data partitioning (e.g., training/validation/testing splits) and preprocessing steps beyond denoising are not clearly detailed. Reproducibility would be improved by explicitly stating the data split ratios, any normalization/scaling techniques, and how missing data (if any) were handled.
2. Minor grammatical issues and occasional typographical errors reduce the manuscript’s polish. Professional language editing or proofreading would enhance clarity and readability.

---

## Round 0.2 · accepted · Accept

The authors have adequately addressed the reviewers' comments and revised the manuscript accordingly. I recommend the manuscript for acceptance and publication.

·

Basic reporting

1. In Table 1, what does 'individual modeling' mean? Instead of using a cross mark (×), please provide a description.

2. Improve the figure quality throughout the manuscript

Experimental design

The authors have incorporated all the suggestions.

Validity of the findings

The authors have incorporated all the suggestions.

·

Basic reporting

The manuscript is written in clear, unambiguous, and technically correct English, conforming to professional standards of academic writing. The Introduction provides sufficient context and clearly articulates the motivation for the study.

Experimental design

The Methods section provides sufficient detail to enable replication, including information on the code, dataset, and computing infrastructure.

Validity of the findings

The formal results section includes precise definitions of terms and, where applicable, provides detailed and logically sound proofs.

Additional comments

The manuscript meets the expectations for clarity, rigor, and scholarly presentation.